# Bis(pyridine)enaminone as a Precursor for the Synthesis of Bis(azoles) and Bis(azine) Utilizing Recent Economic Green Chemistry Technology: The Q-Tube System

**DOI:** 10.3390/molecules28052355

**Published:** 2023-03-03

**Authors:** Khadijah M. Al-Zaydi, Tamer S. Saleh, Norah F. Alqahtani, Meaad S. Bagazi

**Affiliations:** 1Department of Chemistry, College of Science, University of Jeddah, Jeddah 21589, Saudi Arabia; 2Green Chemistry Department, National Research Centre, Dokki, Giza 12622, Egypt

**Keywords:** Q-Tube reactor, green chemistry, bis(pyridine enaminone), bis(azoles), bis(azines)

## Abstract

We reported herein efficient economic high-pressure synthesis procedures for the synthesis of bis(azoles) and bis(azines) by utilizing the bis(enaminone) intermediate. Bis(enaminone) reacted with hydrazine hydrate, hydroxylamine hydrochloride, guanidine hydrochloride, urea, thiourea, and malononitrile to form the desired bis azines and bis azoles. A combination of elemental analyses and spectral data was used to confirm the structures of the products. Compared with conventional heating, the high-pressure Q-Tube method promotes reactions in a short period of time and provides high yields.

## 1. Introduction

Temperatures can easily be controlled in the range essential for chemical reactivity, which is probably why organic chemists prefer temperature over high pressures. On the other hand, high-pressure reactors are the workhorses of chemical industries. They can run chemical reactions at pressures of a few thousand atmospheres [1]. Since chemical reactions are influenced by pressure [2], by using pressure, chemists can force chemical reactions to occur and accelerate transitions between solids, liquids, and gases. Initially, we need to focus on the reaction volume and volume of activation in order to understand pressure effects [2]. The activation volume represents a change in volume from the reactant to transition state, and the reaction volume represents a change in volume from the reactant to the product [3]. According to the reported literature that dealt with this issue [3,4], an illustration of bond formations in Figure 1 schematically clarifies the working concept.

High-pressure reactions with a positive activation volume or reaction volume are expected to occur more slowly than at low pressure, as the positive activation volume or reaction volume works against compression. Therefore, we should carefully choose the reaction that will occur under pressure [5]. For example, during some bond cleavage phenomena, the transition state has a larger volume than the reactant, because it is less compact, and vice versa for bond formation. An S_N_1 reaction mechanism (first step), for instance, showing a significantly negative activation volume might seem counterintuitive because a bond breaks and ionizes (or separates) when charges occur. As such, the cleavage of heterolytic bonds does contribute to the positive activation volume, but the contribution of the volume of ions formed with solvent molecules should not be neglected in an overall volume change, which finally produces a negative activation volume. A very important note was reported [6] that led to a useful summary of the activation volume (positive or negative) for the most common reaction, which helped us decide whether to use high-pressure reaction conditions or not.

A positive activation volume was recorded for the following reactions: neutralization, bond cleavage, charge dispersal, and diffusion control. On the other hand, the following reactions showed a negative activation volume: ionization, cyclization, bond formation, displacement, steric hindrance, and charge concentration. 

Using high-pressure chemistry, solvents, reagents, and final products can be improved in terms of their physical properties in order to achieve quicker and cleaner transformations. The most modified properties are the physical properties of liquids. A pressure reaction has the advantage of overcoming the solvent’s boiling point [7,8,9]; solvent boiling points are directly proportional to vapor pressure when pressure increases. As defined by the Arrhenius equation, the reaction rate can double relative to every 10 °C increase in temperature [10]. A Q-Tube device for high-pressure chemistry was developed, and it is undoubtedly the most straightforward and cheapest alternative to expensive microwave synthesizers (Figure 2).

Q-Tube, unlike other pressure reactors, features a pressure release and resealing system to prevent overpressure explosions [11]. Of note, the conversions and yield values of the Q-Tube high-pressure reactor are higher than that of the MW reactor [12]. Q-Tube influences two factors that enhance the organic reaction. One of them is carried out at an extreme temperature that exceeds the solvent’s boiling point. A temperature increase of 10 °C doubles the reaction rate, according to the Arrhenius equation, as represented in Equation (1). The other factor is an increase in the concentration of substrates due to the high pressure exerted by the Q-Tube system that reduces the reaction volume, including liquids; consequently, the collision frequency increases, leading to faster reactions.
(1)$$k=Ae^{-Ea/RT}$$

A rate constant is defined as *k*, activation energy is defined as *Ea*, and a gas constant is defined as *R* (8.3145 J/Kmol). In contrast, temperature is defined as *T*. A frequency factor of *A*, with units of L∙mol^−1^∙s^−1^, is calculated by considering the frequency of reactions and the likelihood of correct molecular orientation [13,14]. On the other hand, pyridine rings attracted particular attention regarding their unique biological activities. It has been claimed that many pyridine derivatives possess engaging biological and pharmacological properties, such as antimicrobial [15], antitubercular [16], anticonvulsant, analgesic, anti-inflammatory [17], and anticancer properties [18,19,20]. In addition, one of the most elementary classes that exhibit potential biological and pharmacological activities is bis-heterocyclic compounds [21,22,23]. Moreover, enaminones are valuable intermediates for synthesizing several heterocyclic systems in addition to their pharmacological potentialities [24,25,26]. Aiming to obtain bis-pyridine enaminone as a precursor for different pyridine-azoles hybrids and pyridine-azine hybrids via green synthesis for biological screening purposes, several studies show that pyridine–azole hybrids and pyridine–azine hybrids have fascinating medicinal properties [27]. These hybrids show the highly potent and selective mGlu5 receptor antagonist with good brain penetration and receptor occupancy [28] as well as calcium channel blocking for the treatment of hypertension [29]. Moreover, they exhibited excellent antifungal activity against *Candida albicans* [30] and inhibited bacterial biofilms, DNA gyrase, and *Mycobacterium tuberculosis* [31]. Moreover, they demonstrated antihyperglycemic activity [32], in which their compounds metabolically stabilized the glucokinase activator [33] and exhibited acute oral glucose-lowering efficacy. In addition, some hybrids showed antiviral activities as potent efficacy against human cytomegalovirus (CMV) and the varicella-zoster virus [34]. For cancer treatments, several hybrids showed cytotoxicity against several human cancer cells, including ovarian cancer cells [35] and gastric cancer cells [36], with antiproliferative effects [37]. Green chemistry is based on the design (or redesign) of products and/or manufacturing processes to reduce their impact on human health and the environment [38,39]; therefore, the green organic synthesis trend induces the prevention of the harmful impact on the environment. Catalysis, and any system that reduces reaction time to save energy and lower emissions, is an excellent choice by applying different green chemistry principles [38,39,40]. Our previous results encouraged us to continue synthesizing biologically active heterocyclic compounds [41,42,43,44,45,46] using other green chemistry methods [47,48,49,50,51,52,53,54,55,56,57]. A high-pressure Q-Tube reactor is used in this study to synthesize bis(enaminone), an essential building block for synthesizing bis-heterocyclic compounds.

## 2. Results and Discussions

The design of the Q-Tube™ system allows it to be operated safely and accurately with high reproducibility. The design is clarified in Figure 3, showing the presence of a Qian cap (Figure 3a), which consists of a Teflon septa and needle (Figure 3b) to release pressure via the perforation of the Teflon septa (acts as active ventilation for safety, avoiding a rapid increase in pressure). 

The reactivity of enaminone towards different nucleophiles is frequently studied by many reports [58,59]. Figure 1, Figure 2 and Figure 3 show the synthetic procedures used to obtain the target compounds. In a reflux and/or Q-Tube reactor, we used *N*,*N*-dimethylformamide dimethyl acetal (DMF-DMA) **2** to react with 2,6-diacetylpyridine **1** in the presence of a little excess of DMF-DMA to produce a single product identified as bis(pyridine)enaminone **3** (Figure 1). A high percentage of yield was observed when using high-pressure reactors (Q-Tubes) instead of the conventional reflux method [60,61]. With the previously reported microwave radiation method, almost the same % yield was obtained utilizing high-pressure reactors (Q-Tubes) [62]. Figure 1 and Table 1 show comparative results between the two preparation methods. 

The analytical data for both preparation methods agreed with structure **3** (Figure 1). The ^1^H spectrum of the isolated compound exhibited two singlet signals at δ 2.90 and 3.20 due to *N*,*N*-dimethylamino protons, two doublet signals at δ 6.55 and 7.85 (*J =* 12.3 Hz) due to olefinic protons, and pyridine ring protons showed two doublet signals at δ 8.074 ppm and a triplet signal at δ 8.012 ppm. Based on the olefinic proton coupling constant value of 12.3 Hz, bis(enaminone) **3** has an *E*-configuration [63]. By condensing hydrazine hydrate and hydroxylamine with bis(enaminone), we investigated its potential for the synthesis of novel bis(pyrazoles) and bis(isoxazoles). Compound **3** reacted with hydrazine monohydrate in ethanol in the high-pressure reactor Q-Tube. A single product was identified as bis(pyrazole) derivative **4** (Figure 2). There are two doublets identified as one doublet for CH-4 pyrazole at δ 6.95 ppm and one doublet for CH-5 pyrazole at δ 7.62 ppm in its ^1^HNMR spectroscopy [64].

By using the high-pressure reactor Q-Tube, compound **3** also condensed with hydroxylamine hydrochloride in the presence of a base such as sodium carbonate in ethanol as a solvent. Bis(isoxazole) **5** was detected exclusively in TLC. ^1^HNMR indicated that two characteristic doublets were obtained for the isolated compound, with one of them at δ 8.78 (d, *J =* 1.86 Hz CH-5 isoxazole) downfield while the other was at δ 7.25 (d, *J =* 1.86 Hz CH-4 isoxazole) upfield [65], which is completely consistent with the proposed structure depicted in Figure 2. It is noteworthy that to find an advantage of the high-pressure reactor on the reactions shown in Figure 2, these reactions had been carried out with conventional heating; for both compounds **4** and **5**, almost the same yield but shorter reaction times was found with the high-pressure Q-Tube reactor. Figure 2 and Table 2 represent the time of reactions and % yield.

The Q-Tube reactor demonstrated that bis(enaminone) **3** is an important precursor in synthesizing bis(azines) **6**–**8** from its reactions with guanidine, urea, thiourea, and malononitrile; only one product was obtained in each case (Figure 3). By utilizing the high-pressure Q-Tube in comparison to conventional heating (Figure 3), bis(enaminone) **3** gives excellent yields and the fastest reaction times for pyrimidines resulting from reactions with guanidine hydrochloride, urea, and/or thiourea. The IR spectrum of compound **6** shows symmetric and asymmetric bands at 3250–3450 cm^−1^ due to the amino group [64]. Its ^1^HNMR shows a D_2_O exchangeable singlet peak at δ 6.79, representing the amino group [65].

The IR spectrum of compound **7a** exhibits an OH absorbance band at 3284 cm^−1^ [64], which confirms the presence of tautomeric isomer “enol form is predominant”, as shown in Figure 4. Moreover, ^1^HNMR confirms the presence of the D_2_O exchangeable signal for OH due to two Hydroxy groups at 5.89 ppm (cf. Materials and Methods).

Compound **7b** shows a distinct absorbance band due to C=S in IR at 1407 cm^−1^ and a signal in ^13^CNMR above 185 ppm for (C=S). As tested by TLC, only one compound formed from the reaction of compound **3** with malononitrile in ethanol and drops of piperidine at 120 °C/25 psi. A band at 2174 cm^−1^ is evident in the IR spectrum of the formed compound **8** due to the cyano group, and the ^1^HNMR spectrum is consistent with the formed mixture (cf. Materials and Method). The obtained spectroscopic data confirm that compound **8** was identified as bis(pyridine) via the pyridine bridge (Figure 3). According to Table 3, the compounds formed using the high-pressure Q-Tube have a higher yield and a shorter reaction time than the conventional heating method.

Of note, the formation of compound **8** is in line with the reported literature [66], which suggested that malononitrile substitutes the dimethylamino group first to give intermediate A, followed by cyclization to bisaminopyran (intermediate B) and then the ring opening of the formed aminopyran to afford intermediate C, which readily eliminates dimethylamine to afford the final isolable pyridinone derivative **8** (in a Dimroth-type rearrangement) (Figure 4).

Finally, we succeeded in introducing a technique considered as state of the art in modern green organic chemistry techniques. Green chemistry tends to perform organic reactions under atmospheric pressures, avoiding the high amount of energy that is consumed to obtain highly pressurized reactions, or develops a new catalyst to reduce the operating temperature and pressure for the process; consequently, less energy is consumed, which is good for the industry. However, the Q-Tube system works in a manner that does not consume more energy to attain high pressure as the pressure released in our reactions described above is considered autogenic pressure. Another important advantage is the high reproducibility of the obtained results. On the other hand, we succeeded in applying the use of this economic Q-Tube system in comparison to the microwave reactor in order to synthesize the bis(azoles)-pyridine hybrids and bis(azines)-pyridine hybrids of expected high biological and pharmacological activities.

## 3. Materials and Methods

### 3.1. General

All melting points were measured on a Gallenkamp Electrothermal melting point apparatus (Gallenkamp, Cambridge, UK) and were uncorrected. The infrared spectra (KBr disks) in 4000 to 400 cm^−1^ were recorded on a Perkin-Elmer Frontier spectrometer (Perkin-Elmer, San Diego, CA, USA). The N.M.R. spectra were recorded on a 850 and 600 MHz NMR spectrometer (Bruker, Fällanden, Switzerland) that is deuterated in dimethylsulphoxide (DMSO-*d*_6_) and chloroform (CDCl_3_). Chemical shifts were quoted in δ and were related to that of the solvent. Mass spectrometry was carried out using a direct probe controller inlet part in a single quadrupole mass analyzer (Thermo Fisher Scientific GCMS Model I.S.Q. LT, Carlsbad, CA, USA) using Thermo X-Calibur Software at the regional center for mycology and biotechnology (R.C.M.B.), Al-Azhar University, Cairo. The reaction temperature was manually input depending on the boiling point of the solvent used and stabilized for more than an hour. Q-Tube-assisted reactions were performed in a Q-Tube-safe pressure reactor from Q Labtech (Washington, DC, USA) equipped with a cap/sleeve, pressure adapter (120 psi), needle, borosilicate glass tube, Teflon septum, and catch bottle. Elemental analyses were performed using a Perkin-Elmer 2400 Analyzer (Perkin-Elmer, San Diego, CA, USA). T.L.C. Sigma-Aldrich (St Louis, MA, USA) silica gel was used on TLC Al foils and a silica gel matrix, with a fluorescent indicator at 254 nm. The ^1^HNMR spectra of the synthesized compounds can be found in Appendix A

### 3.2. General Procedure for Preparation of Dienaminone ***3***

Method I: A mixture of compound **1** (10 mmol) and DMF-DMA (25 mmol) was refluxed at 90–100 °C overnight. The mixture was left at room temperature, and the solid was collected by hexane and petroleum ether 40–60 and then washed by ethanol.

Method II: The same above mixture scale was placed in a Q-Tube at 120 °C/under the autogenic pressure (15 psi) for 23–24 min. The products were collected and washed with ethanol: one spot on TLC; eluent (CHCl_3_:MeOH 85:15).

#### *2,6-Bis(3-dimethylamino-1-oxoprop-2-en-yl)pyridine* (**3**) [62]

Orange shine grain; (Q-Tube) yield 94%; m.p. 236–237 °C, FT-IR cm^−1^: 3026, 2911 (CH aromatic), 2807(CH aliphatic), 1643(C=O ketone); fingerprint area matched precisely with the product obtained from conventional heating. ^1^HNMR (850 MHz, DMSO-*d*6): δ, ppm = 2.96 (s, 6H), 3.20 (s, 6H), 6.55 (s, 2H broad alkene), 7.84–7.85(d, 2H alkene), 8.01–8.07 (dt, 3H, pyridine ring).

### 3.3. General Procedure for Preparation of Bis(azoles) ***4*** and ***5***

Method I: For bis(pyrazoles), A mixture of compound **3** (10 mmol) and hydrazine monohydrate (20 mmol) in ethanol was refluxed at 50–60 °C for 120 min. The mixture was left to cool and then added to iced water. The solid was collected by filtration and washed by EtOH. For bis(isoxazoles), a mixture of compound **3** (10 mmol), hydroxylamine hydrochloride (25 mmol) and sodium carbonate (25 mmol) in EtOH was refluxed at 50–60 °C for 240 min. The mixture was left to cool and then added to iced water. The solid was collected by filtration and washed by EtOH. 

Method II: The same above mixture scale was placed in a Q-Tube at 120 °C/under an autogenic pressure of 30 psi for an appropriate period time as examined by TLC. Then, the products were obtained by following the same workup mentioned above.

#### *2,6-Bis(1H-pyrazol-3-yl)pyridine* (**4**) [62]

White cotton; (Q-Tube) yield 75%; m.p.c 257–259 °C, FT-IR cm^−1^: 3190; 3049; 2947 (CH aromatic), 1591 (C=N), 1596–1564 (C=C aromatic). ^1^HNMR (850 MHz, DMSO-*d*_6_): δ ppm = 6.95 (d, 2H, CH-4 pyrazole), 7.62−7.86 (m, 2H, CH-5 pyrazole and 3H pyridine ring), 13.02 (s, 1H, NH), 13.51 (s, 1H, NH).

#### *2,6-Di(isoxazol-5-yl)pyridine* (**5**) [62]

Beige powder; (Q-Tube) yield 29%; 143–147 °C, FT-IR cm^−1^: 3126, 3097 (CH aromatic), 1601 (C=N). ^1^HNMR (600 MHz, DMSO-*d*_6_): δ, ppm = 7.25 (d, 2H, CH-4 isoxazole), 8.76 (d, 2H, CH-5 isoxazole), 8.08 (d, 2H, pyridine), 8.19 (t, 1H, pyridine).

### 3.4. General Procedure for Preparation of Bis(pyrimidines) ***6***, ***7a*** and ***7b***

Method I: A mixture of compound **3** (10 mmol), guanidine hydrochloride, urea or thiourea (20 mmol), and sodium ethoxide (30 mmol) in EtOH was refluxed at 60–70 °C for the appropriate time, as examined by TLC. The mixture was left to cool. The solid was collected by filtration and washed by EtOH.

Method II: The same above mixture scale was placed in a Q-Tube at 120 °C under an autogenic pressure of 30 psi for an appropriate period of time as examined by TLC. The products were collected and washed by ethanol. Recrystallization by DMF.

#### *4,4′-(Pyridine-2,6-diyl)bis(pyrimidin-2-amine)* (**6**) [67]

Pink powder; (Q-Tube) yield 97%; m.p. > 300 °C, FT-IR cm^−1^: 3442; 3290 (NH2), 3145 (CH aromatic), 1596 (C=N), 1549 (C=C aromatic), ^1^HNMR (850 MHz, DMSO-*d*_6_): δ, ppm = 6.79 (4H, NH2), 7.65 (d, 2H, CH-5 pyrimidine), 8.40 (d, 2H, CH-6 pyrimidine), 8.46 (d, 2H, pyridine), 8.16 (t, 1H, pyridine).

#### *4,4′-(Pyridine-2,6-diyl)bis(pyrimidin-2-ol)* (**7a**) [67]

Dark violet powder; (Q-Tube) yield 96% m.p. > 300 °C, FT-IR cm^−1^: 3284 (OH), 1575 (C=C aromatic). ^1^HNMR (850 MHz, DMSO-*d*_6_): δ, ppm = 5.89 (d, OH), 7.69–8.51 (m, 7H, pyridine and pyrimidine) 9.49 (d, OH). GC-MS: *m*/*z* [M]^+^ 267.84 

#### *4,4′-(Pyridine-2,6-diyl)bis(pyrimidine-2(1H)-thione)* (**7b**) [67]

Orange powder; (Q-Tube) yield 41%, m.p. > 300 °C. FT-IR cm^−1^: 3306 (NH), 1532; 1556 (C=C aromatic), 1605 (C=N); ^1^HNMR (600 MHz, DMSO-*d*_6_): δ, ppm 7.52 (d, 2H, pyrimidine ring) 8.05 (t, 1H, pyridine ring) 8.16 (d, 2H, pyridine ring) 8.38 (d, 2H, pyrimidine ring); ^13^CNMR (213 MHz, DMSO-*d*_6_): δ, ppm = 127.3, 135.4, 141.0, 158.7, 179.2, 194.8. GC-MS: *m*/*z* [M]^+^ 299.27.

### 3.5. General Procedure for Preparation of Bis(pyridine)derivative ***8***

Method I: A mixture of compound **3** (10 mmol) and malononitrile (20 mmol) and piperidine drops in EtOH was refluxed at 75 °C for 1440 min. The solid was collected by filtration and washed by EtOH. 

Method II: The same above mixture scale was placed in a Q-Tube at 120 °C under the autogenic pressure of 30 psi for an appropriate period of time, as examined by TLC. The product was collected and washed by ethanol, Recrystallization by EtOH.

#### *6,6″-Dioxo-1,1″,6,6″-tetrahydro-[2,2′:6′,2″-terpyridine]-5,5″-dicarbonitrile* (**8**) [68]

Orange powder; (Q-Tube) yield 99%, m.p. 202–204 °C; FT-IR cm^−1^: 3412; 3326 (OH and NH keto-enol form), 2184 (C≡N), 1645 (C=O) and 1534 (C=C aromatic); ^1^HNMR (850 MHz, DMSO-*d*_6_): δ, ppm = 5.62–5.64 (dd, 1H), 6.09–6.11 (dd, 1H), 6.99–7.04 (dd, 2H), 7.48–7.53 (dd, 2H), 8.11 (t, 1H, pyridine), 6.85 (broad s, OH), 6.69 (s, OH).

## 4. Conclusions

This study developed an efficient green organic synthesis protocol that saves energy by using a high-pressure Q-Tube reactor as an economic and safe alternative to a microwave reactor. A bis(enaminone) precursor was used in this protocol for preparing bis azoles and bis azines of expected biological and pharmacological efficacy. A mechanism for the reaction of bis(enaminone) with malononitrile suggested that the reaction proceeded via a Dimorth-type rearrangement. Modern Q-Tubes with a Qian cap have the fastest reaction time compared to conventional heating methods and almost identical results to microwave reactors. The technique opens the door to working on new substrates for biological screening in the future.

## Data Availability

The data presented in this study are available in Appendix A and upon request from the corresponding author.

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
