# Peer review of "Bis(pyridine)enaminone as a Precursor for the Synthesis of Bis(azoles) and Bis(azine) Utilizing Recent Economic Green Chemistry Technology: The Q-Tube System"

_molecules, 2023, doi:10.3390/molecules28052355_

Round 1

Reviewer 1 Report

The authors described an efficient synthesis of bis(azoles) and bis(azines) utilizing the bis(enaminone) intermediate in the high-pressure Q-tube. Compared with conventional heating, the high-pressure Q-tube method promotes the reactions in a short time and with high yields. I think it is very useful and practice methodology, so I recommend this paper should be accepted by Molecules after minor revision.

The ratio of bis(pyridine)enaminone 3 with various nucleophilic reagent is 1 : 2, but the yields of products 4, 5 and 7b were unsatisfactory, so I suggest the authors should test in high ratio of them. Moreover, the side-product might be formed during the reaction of bis(pyridine)enaminone 3 with thiourea. 

Author Response

Dear Reviewer 1

Thank you very much for your valuable comments which add to our manuscript, Please find enclosed the attached file of the response to your esteemed comments 

Tamer S. Saleh

Reviewer 2 Report

Review Report: molecules-2172603

Title: Bis(pyridine)enaminone as a precursor for the synthesis of 2bis(azoles) and bis(azine) utilizing recent economic green chemistry technology: The Q-Tube system

The present work is of good quality and is a unique effort in its form to utilize a nonconventional Q-tube system for synthesizing bipyridine derives through greener route. Before its final publication, following few points may be undertaken by the authors

·         Breathing conditions of Q-tube systems is missing please explain it with proper citation

·         Q-tube system is a non-conventional technique for greener synthesis of organic compounds at nanoscale, however no discussion is provided about the size morphology of compounds.

·         A comparison with conventional heating strategies and unconventional energy source such as microwave or ultrasounds assisted synthesis may be provided by the current and previous studies.

·         Introduction should be enriched with the importance of Bis(pyridine)enaminone, 2 bis(azoles) and bis(azine) derivatives. First paragraph should also highlight the scope of green chemistry.

·         Results and discussion is too ambiguous to understand. I will suggest to bifurcate it in different section (IR, NMR etc)

·         No mass spectra is provided

·         Line 89: bis(isoxazole) 5 was detected exclusively in TLC… Please explain it.

·         Line 116: The IR spectrum of compound 7a exhibits OH absorbance band at 3284cm-1, which confirms the presence of tautomeric isomer "enol form is predominant," …Please explain how the type and nature of IR spectral peak can distinguish between two different isomeric forms.. Also no IR spectra are given in supplementary information. Please provide all the IR spectra in supplementary information and at least one IR spectra as a representative in the main text manuscript.

·         All the spectral data of compound may be shifted from supplementary information to main text manuscript.

·         1HNMR spectral peaks are wrongly quoted in main text (like amine at 6.72).. Please check them all and correct them accordingly.. Representative NMR spectrum may be provided in main text.

·         All the experimental procedures/protocol and materials should be present in main manuscript with their proper citations.

·         cm-3 should be cm-1, 1HNMR should be 1HNMR and D2O should be D2O.. Please check all the manuscript thoroughly

·         Conclusions are nice but very short please rewrite them.

Author Response

Dear Reviewer 2

Thank you very much for your valuable comments, which add to our manuscript. I am sharing with you the attached file of the response to your esteemed comments 

Tamer S. Saleh

Reviewer 3 Report

A manuscript by Khadijah M. Al-Zaydi et al. describes the chemistry of unsaturated ketones to produce heterocyclic compounds under optimized conditions. It should be noted that the chemistry of substituted unsaturated ketones is well known and the novelty of this work is the modification of the conduct of condensations. 

During the reading some remarks arose. Firstly, the authors should mention the review articles on enaminoketones and alkoxyenones, which show the main conditions and possibilities for the formation of heterocyclic compounds (https://doi.org/10.1002/ejoc.202001039; https://doi.org/10.1070/RC2014v083n02ABEH004388). Based on these references, we can assume that the structure of compound 5 is not correctly shown. The addition of hydroxylamine should occur first of all at the activated double carbon-carbon bond, and then the intramolecular cyclization of the hydroxyl group along the carbonyl fragment is realized. How did the authors confirm the regioselectivity of the formation of product 5? 

The next question relates to product 8. It is necessary to include in the text of the article an explanation of how this transformation occurs (suggested mechanism). Otherwise, it is necessary to provide literature references in which this is discussed in detail. 

Author Response

Dear Reviewer 3

Thank you very much for your valuable comments, which add to our manuscript. I am sharing with you the attached file of the response to your esteemed comments 

Tamer S. Saleh

Round 2

Reviewer 2 Report

Review Report: Molecules-2172603R

Authors addressed all the comments partially. Authors didn’t able to understand few points. I understand that the authors didn’t work on nano scale but why they used a system which is suitable for synthesizing compounds at nanoscale?. Authors are supposed to elaborate their working design/protocol with its working specification as this new system may be adopted by new synthetic scientist. They can take help from recently published research as well. What was the rationale for it; need to be discussed in paper?

They are strongly advised to describe the breathing conditions of Q-tube system in present study. 

All the discussion of IR and NMR spectra is without any citation and any discussion. For better understanding every cited peak must be supported with a citation. Please cite the following papers with the peak to make a comparison. For better understanding they can cite the recently published relevant papers such as

doi.org/10.1016/j.molstruc.2021.131710

doi.org/10.1002/aoc.6054

New addition of paragraphs into introduction is also not good. Each line should be supported with proper citation. Rather than to enrich it with its prospects, uses and efficiency, they are narrating their aims and objectives. Please rewrite them.

Quality of figure 3 is extremely bad and is impossible for anyone to read. 

Fig 2 and Fig 3 may be merging into single one.

A careful read can decide that there are still many typos (like  LiTaO3). Please check them.

No any effort is made to improve the conclusions as suggested in previous round.

Author Response

Reviewer 2 Comments

Authors addressed all the comments partially. Authors did not able to understand few points.

  1. I understand that the authors didn’t work on nano scale but why they used a system which is suitable for synthesizing compounds at nanoscale? Authors are supposed to elaborate their working design/protocol with its working specification as this new system may be adopted by new synthetic scientist. They can take help from recently published research as well. What was the rationale for it; need to be discussed in paper?

Response

Thank you very much for your esteemed comment and your valuable clarification, now we understand your comment in which we undergo the organic reaction under pressure which like preparation of nanomaterial. Just we clarify here, always synthetic organic chemist search for new tools to attain the green chemistry principles such decrease reaction time (that leads to decrease consumed energy which considered a very important principle in green chemistry) and increase the yield of products. As we know that most of companies that design the microwave reactor for synthesis such as CEM™, Anton paar™ and Mielstonse™ developed a technique for organic synthesis based autogenic pressure with active ventilation system (to avoid explosion) in order to decrease the reaction time of organic synthesis not only based on microwave energy but also on pressure to decrease reaction time consequently decrease the energy consumed according to Arrhenius equation that already depicted in our manuscript. On the other hand, Q tube system based on autogenic pressure that generated in situ from solvent evaporation, the main advantage of it is economic in comparing with microwave reactor, in addition no limitation on solvents used. Therefore, it consider a good economic replacement for microwave synthesizer, a pressure / sealed tube, a round bottom flask/condenser or a hydrogenation devices. Noteworthy, the hydrothermal autoclave that used in nano scale synthesis not easy to use in organic synthesis in which no ventilation that may cause burning of organic compounds and formation of carbon dots. Also, the Q tube synthesis provide us with the safety not as normal pressure tube that may be exploded as the Q tube is connected with Qian cab (described in details in your esteemed comments 2). Also, many research paper for different worldwide research group utilized this patented system in organic synthesis as in the following references as example:

References:

·         Regioselective ketone α-alkylation with simple olefins via dual activation. Fanyang Mo, Prof. Guangbin Dong, Science 4 July 2014, Vol. 345 no. 6192 pp. 68-72.

·         MCRs for the synthesis of 2,3-dihydroquinazolin-4(1H)-ones, Luca Sancineto, Bonifacio Monti, Orsola Merlino, Ornelio Rosati, and Claudio Santib. Arkivoc 2018, part III, 270-278.

  • Organic Reactions under High Pressure: Efficient Multicomponent Synthesis of Novel Tricyclic Pyridazinonaphthyridine Derivatives under High Pressure, Moustafa, Moustafa S.; Al-Mousawi, Saleh M. Current Organic Chemistry, Volume 22, Number 3, January 2018, pp. 268-275(8).
  • Oxidation of aromatic oxygenates for the production of terephthalic acid, Konstantinos A.Goulas, Mika Shiramizu, James R.Lattner, Basudeb Saha, Dionisios G.Vlachos. Applied Catalysis A: General, Volume 552, 25 February 2018, Pages 98-104.
  • Q-tube System, A Nonconventional Technology for Green Chemistry Practitioners Nacca, Francesca G.; Merlino, Orsola; Mangiavacchi, Francesca; Krasowska, Dorota; Santi, Claudio; Sancineto, Luca. Current Green Chemistry, Volume 4, Number 2, August 2017, pp. 58-66(9).
  • Mechanism and kinetics of 1-dodecanol etherification over tungstated zirconiaJulie Rorrer, Ying He, F. Dean Toste, Alexis T.Bell. Journal of Catalysis, Volume 354, October 2017, Pages 13-23.
  1. They are strongly advised to describe the breathing conditions of Q-tube system in present study. 

Response

Thank you for your valuable comment, the best clarification for ventilation (breathing) in Q-tube in the following figures in which Q Tube is provided with system called Qian Cab™ this is for safe release of excess pressure which based on Teflon septum that can be perforated with needle in case of high pressure as declared in the figures below

(c)                                     (b)                                              (a)

Figure A is Qian cap system (represented by blue circle), Figure b schematic diagram represent the reaction in Q-tube system with Teflon septum (orange color), Figure C represented in case of high pressure the Teflon septa is expanded to above so will meet the needle that perforated it and breathing occur as

Let me do not put these details in manuscript in which I already put reference for it and I select part of it to included in manuscript.

  1. All the discussion of IR and NMR spectra is without any citation and any discussion. For better understanding every cited peak must be supported with a citation. Please cite the following papers with the peak to make a comparison. For better understanding they can cite the recently published relevant papers such as doi.org/10.1016/j.molstruc.2021.131710, doi.org/10.1002/aoc.6054

Response

Thank you for your valuable comment, we cited the mentioned papers for comparison.

  1. New addition of paragraphs into introduction is also not good. Each line should be supported with proper citation. Rather than to enrich it with its prospects, uses and efficiency, they are narrating their aims and objectives. Please rewrite them.

Response

Thank you for your valuable comment, we cited the proper citation for support.

  1. Quality of figure 3 is extremely bad and is impossible for anyone to read. Fig 2 and Fig 3 may be merging into single one.

Response

Thank you for your valuable comment, we modify the two figures with high resolution but I see no need for merging them.

  1. A careful read can decide that there are still many typos (like  LiTaO3). Please check them.

Response

Thank you for your valuable comment, we revised all typo errors as possible.

  1. No any effort is made to improve the conclusions as suggested in previous round.

Response

Thank you for your valuable comment, we tried to improve the conclusion as possible  in which we include in it all the important results.

Reviewer 3 Report

The authors have amended the revised version of the article according to the comments of the reviewers. On this basis, I can recommend this manuscript for publication.

Author Response

Reviewer 3 comments

The authors have amended the revised version of the article according to the comments of the reviewers. On this basis, I can recommend this manuscript for publication.

Thank you very much for your esteemed decision.
